# The Role of CXCR1, CXCR2, CXCR3, CXCR5, and CXCR6 Ligands in Molecular Cancer Processes and Clinical Aspects of Acute Myeloid Leukemia (AML)

**DOI:** 10.3390/cancers15184555

**Published:** 2023-09-14

**Authors:** Jan Korbecki, Patrycja Kupnicka, Katarzyna Barczak, Mateusz Bosiacki, Paweł Ziętek, Dariusz Chlubek, Irena Baranowska-Bosiacka

**Affiliations:** 1Department of Biochemistry and Medical Chemistry, Pomeranian Medical University in Szczecin, Powstańców Wlkp. 72, 70-111 Szczecin, Poland; jan.korbecki@onet.eu (J.K.); patrycja.kupnicka@pum.edu.pl (P.K.); mateusz.bosiacki@pum.edu.pl (M.B.); dchlubek@pum.edu.pl (D.C.); 2Department of Anatomy and Histology, Collegium Medicum, University of Zielona Góra, Zyty 28, 65-046 Zielona Góra, Poland; 3Department of Conservative Dentistry and Endodontics, Pomeranian Medical University in Szczecin, Powstańców Wlkp. 72, 70-111 Szczecin, Poland; katarzyna.barczak@pum.edu.pl; 4Department of Orthopaedics, Traumatology and Orthopaedic Oncology, Pomeranian Medical University, Unii Lubelskiej 1, 71-252 Szczecin, Poland; pawel.zietek@pum.edu.pl

**Keywords:** leukemia, AML, chemokine, interleukin-8 (IL-8), interferon-gamma-inducible protein 10 kDa (IP-10), bone marrow, blood, macrophage migration inhibitory factor (MIF)

## Abstract

**Simple Summary:**

AML is a type of leukemia with a very unfavorable prognosis. Some of the new therapeutic targets that are being investigated by researchers worldwide are chemokines of the CXC subfamily, which includes CXCL12. Although this chemokine has been very well studied, other CXC chemokines has been less frequently examined in AML. There is also a lack of a review summarizing the role of CXC chemokines other than CXCL12 in AML. For this reason, this review describes the significance of the ligands for receptors CXCR1, CXCR2, CXCR3, CXCR5, and CXCR6 in AML. The focus is on clinical aspects as well as molecular cancer processes in AML.

**Abstract:**

Acute myeloid leukemia (AML) is a type of leukemia known for its unfavorable prognoses, prompting research efforts to discover new therapeutic targets. One area of investigation involves examining extracellular factors, particularly CXC chemokines. While CXCL12 (SDF-1) and its receptor CXCR4 have been extensively studied, research on other CXC chemokine axes in AML is less developed. This study aims to bridge that gap by providing an overview of the significance of CXC chemokines other than CXCL12 (CXCR1, CXCR2, CXCR3, CXCR5, and CXCR6 ligands and CXCL14 and CXCL17) in AML’s oncogenic processes. We explore the roles of all CXC chemokines other than CXCL12, in particular CXCL1 (Gro-α), CXCL8 (IL-8), CXCL10 (IP-10), and CXCL11 (I-TAC) in AML tumor processes, including their impact on AML cell proliferation, bone marrow angiogenesis, interaction with non-leukemic cells like MSCs and osteoblasts, and their clinical relevance. We delve into how they influence prognosis, association with extramedullary AML, induction of chemoresistance, effects on bone marrow microvessel density, and their connection to French–American–British (FAB) classification and FLT3 gene mutations.

## 1. Introduction

Acute myeloid leukemia (AML) is a type of leukemia with very unfavorable prognoses [1]. It is estimated that the median survival for AML patients is approximately one year from diagnosis [2,3,4,5]. The worldwide incidence of this type of leukemia stands at approximately 1.5 cases per 100,000 population [6]. However, in Western countries, this incidence is higher, estimated at around 2.4 cases per 100,000 population in Western Europe and North America [6]. The average mortality rate caused by AML globally is around 1.3 cases per 100,000 population, while in Western Europe and North America, it is approximately 2.2 cases per 100,000 population [6]. The proximity of mortality rates to incidence figures underscores the severity of the disease.

Due to the unfavorable outlook, research is being conducted to develop new therapeutic approaches for AML. One such avenue of research involves examining extracellular factors, including chemokines.

CXC chemokines (α-chemokines) belong to a subfamily of chemokines and are chemotactic cytokines responsible for guiding immune system cells (Table 1) [7]. They play an essential role in the functioning of the immune system. α-chemokines possess a conservative CXC motif at the N-terminus, distinguishing them from other types like β-chemokines and δ-chemokines, which have CC and CX3C motifs at the N-terminus, respectively. In humans, there are 16 representatives of α-chemokines: CXC motif chemokine ligand (CXCL)1–17, except CXCL15, which is a mouse chemokine [7]. α-chemokines activate one of the six α-chemokine-specific receptors: CXC motif chemokine receptor (CXCR)1–6. The expression levels of most of these receptors and chemokines in AML cells are closely associated with patient prognosis [8,9,10]. This suggests that α-chemokines play a significant role in AML’s oncogenic processes and might represent potential therapeutic targets for this leukemia. Currently, compounds targeting the CXCR4 receptor are being tested [11,12,13]. However, drugs targeting other α-chemokine axes in AML, in particular, CXCR1, CXCR2, and CXCR3 ligands, have not yet been explored. Additionally, there is no comprehensive review summarizing the significance of α-chemokines, other than the CXCL12–CXCR4 axis, for AML. This review aims to generate interest in these other cytokines within the scientific community.

## 2. CXCR1 and CXCR2 Ligands

### 2.1. Basic Information about CXCR1 and CXCR2 Receptors and Their Ligands

In humans, there are seven chemokines that act as ligands for CXCR2 (CD182) [7,14]:CXCL1, also known as growth-regulated oncogene (Gro)-α, melanoma growth stimulatory activity (MGSA),CXCL2, also known as Gro-β,CXCL3, also known as Gro-γ,CXCL5, also known as epithelial cell-derived neutrophil-activating factor 78 (ENA-78),CXCL6, also known as granulocyte chemoattractant protein 2 (GCP-2),CXCL7, also known as neutrophil-activating protein 2 (NAP-2), encoded by the pro-platelet basic protein (*PPBP*) gene in the form of pro-peptide, which is then proteolytically shortened to connective tissue-activating peptide III (CTAP-III), β-thromboglobulin (β-TG), and CXCL7 [15,16],CXCL8, also known as IL-8, NAP-1, GCP-1.

Additionally, macrophage migration inhibitory factor (MIF) can act as a ligand for CXCR2 [17,18], but it is not classified as a CXC chemokine. Both CXCL6 and CXCL8 can also activate CXCR1 (CD181) at low concentrations [19]. Therefore, this section will also discuss the significance of CXCR1 in AML. On the other hand, the other chemokines mentioned activate CXCR1 only at much higher concentrations than they do CXCR2 [19,20,21].

Among leukocytes, CXCR1 and CXCR2 expression is mainly found on neutrophils [22,23], making the discussed CXCR2 ligands chemoattractants for these cells [24]. CXCR2 expression is also present on basophils [7,25], indicating that CXCR2 ligands can also affect these cells. Additionally, CXCR2 is expressed on endothelial cells, leading the discussed chemokines to possess pro-angiogenic properties [26,27]. CXCR2 ligands play an important role in solid tumor oncogenic processes [28,29] and in AML.

### 2.2. Levels of CXCR2 Ligands in Patients with AML

In patients with AML, the level of CXCR2 ligands in the blood is higher compared to healthy individuals. Specifically, adult AML patients show elevated levels of CXCL1 [30], CXCL8 [31,32,33,34], and MIF [35] in their blood compared to healthy individuals. In particular, adults aged below 65 years have higher levels of CXCL8 in their blood compared to healthy individuals of the same age [36]. Older AML patients, above 65 years, exhibit even higher levels of CXCL8. However, in healthy individuals of the same age, CXCL8 levels are elevated due to the aging process. Therefore, AML patients aged over 65 do not have elevated CXCL8 levels compared to healthy individuals of the same age [36]. The higher levels of CXCL8 in the blood could result from the production of this chemokine by AML cells and the activation of endothelial cells by AML cells [37], leading to an increase in CXCL8 production by endothelial cells.

Additionally, adults with AML exhibit increased production and levels of CXCR2 ligands in the bone marrow. In particular, higher expression of CXCL2, CXCL3, and MIF, as well as CXCL1 and CXCL8 levels, is found in the bone marrow of AML patients compared to healthy individuals [38,39].

### 2.3. Expression of CXCR2 Ligands in AML Cells

In 1/3 of AML patients, AML cells secrete large amounts of CXCL1 [40,41]. However, another study indicates that the expression of these two chemokines in AML cells is low [42]. The expression of *CXCL1* in AML cells may not be related to the French–American–British (FAB) classification [8,10]. Additionally, the expression of CXCL2 in AML cells may be low compared to the expression of other CXCR2 ligands [42]. The expression of *CXCL2* in AML cells may not be related to the Frenc–American–British (FAB) classification [8,10]. The higher expression of CXCL2 in AML cells, especially in cytogenetically normal AML-M5 cells, may be due to the action of GATA-binding protein 2 (GATA2) [43,44]. CXCL2 in these cells enhances GATA2 activation, indicating a positive feedback loop between the two proteins.

Moreover, the expression of CXCL3 in AML cells may be low compared to the expression of other CXCR2 ligands [42]. *CXCL3* expression in AML cells with the FAB M0–M2 phenotype is higher than in AML cells with the FAB M4–M5 phenotype [8,10]. In 1/3 of AML patients, AML cells secrete large amounts of CXCL5 [40,41]. However, another study indicates that the expression of these two chemokines in AML cells is low [42]. The expression of *CXCL5* in AML cells may not be related to the FAB classification [8,10].

AML cells also produce CXCL6, but in quantities 10 times smaller than CXCL5 and CXCL8, and only in less than half of AML patients [40]. The expression of *CXCL6* in AML cells may not be related to the FAB classification [8,10].

The expression of *PPBP* and *MIF* in AML cells has not been extensively studied. However, a screening conducted on the UALCAN portal suggests that the expression levels of *PPBP* and *MIF* in AML cells are significantly higher compared to other CXCR2 ligands [8,10]. The highest expression of PPBP is found in AML cells with the FAB M7 phenotype [8,10].

On the other hand, around 95% of AML patients have AML cells that secrete large amounts of CXCL8 [40,41,42]. AML cells also secrete more CXCL8 than bone marrow mononuclear cells in pediatric AML patients [45]. Nevertheless, another study shows that AML cells express *CXCL8* mRNA but do not secrete CXCL8 [46].

Considering the FAB classification, *CXCL8* expression is highest in AML cells with the FAB M0 phenotype, while it is lowest in AML cells with the FAB M3 and M5 phenotypes [8,10,47]. However, another study shows that in most cases of AML with the FAB M4–M5 phenotype, *CXCL8* is expressed in AML cells, while in AML with the FAB M0–M3 phenotype, *CXCL8* is expressed in only less than 1/3 of cases [48]. However, some studies have not confirmed this association [49].

Analyzing AML cases excluding FAB M3 AML, *CXCL8* expression is higher in AML cells with fms-related receptor tyrosine kinase 3 (*FLT3*) internal tandem duplication (ITD) mutation compared to cases without this mutation [47]. *CXCL8* expression in AML cells may also be higher in cases of translocation t(8;16)(p11;p13) [50]. This translocation leads to the formation of the monocytic leukemia zinc finger protein (*MOZ*)–CREB-binding protein (*CBP*) fusion gene. MOZ-CBP acts with steroid receptor coactivator-1 (SRC-1), leading to increased activation of nuclear factor-κB (NF-κB) and expression of NF-κB-dependent genes, including *CXCL8*. *MIF* expression is highest in AML cells with the FAB M7 phenotype and lowest in AML cells with the FAB M0 phenotype [8,10].

The high expression of individual CXCR2 ligands in AML cells may vary independently among different patients. Production of CXCL1 and CXCL8 in AML cells has been shown to be correlated. However, in other patients, two other CXCR2 ligands, CXCL5 and CXCL6, are correlated with the production of CC motif chemokine ligand (CCL)13, CCL17, CCL22, and CCL24 [40]. This suggests that in these two groups of AML patients, CXCR2 ligands may have the same role in oncogenic processes, but the most crucial CXCR2 ligand depends on the specific case.

### 2.4. The Level of Expression of CXCR1 and CXCR2 Receptors in AML Cells

The expression of CXCR1 on AML cells is very low compared to the expression of other chemokine receptors such as CC motif chemokine receptor (CCR)1, CCR2, and CXCR4 [40]. The expression of this receptor is highest in AML cells with the FAB M5 phenotype [51]. CXCR1 expression on AML cells is higher in medium/high-risk group patients with the mentioned leukemia than in the low-risk group [51].

On the other hand, the level of CXCR2 expression on AML cells is high compared to other chemokine receptors [33]. However, the level of CXCR2 expression does not differ between CD34^+^ AML cells and CD34^−^ AML cells [40]. *CXCR2* expression is highest in AML cells with the FAB M4–M5 phenotype and lowest in AML cells with the FAB M3 phenotype [8,10,51,52]. The expression of CXCR2 is higher in AML cells with *FLT3* gene mutations than in those without this mutation [52]. Additionally, CXCR2 expression on AML cells is higher in medium/high-risk group patients with the mentioned leukemia compared to the low-risk group [51]. The level of CXCR2 expression in AML cells in adult patients is not associated with white blood cell (WBC) counts and bone marrow blast percentages [52].

### 2.5. The Association of the Expression Levels of CXCR1 and CXCR2 Receptors along with Their Ligands with Outcomes for Patients with AML

Higher levels of *CXCR2* receptor expression on AML cells are associated with poorer prognoses for adult patients [51,52]. The same correlation occurs with *CXCR1* expression [53]. Another available study indicates that higher *CXCR1* expression in AML cells is associated with a tendency (*p* = 0.052) for poorer prognoses [51].

Moreover, the level of CXCR2 ligand expression in AML cells is closely related to patient prognoses. According to the UALCAN website (https://ualcan.path.uab.edu, accessed on 10 July 2023), there is a tendency (*p* = 0.069) for poorer prognoses with higher *CXCL1* expression in AML cells [8,10]. Furthermore, patient survival analysis on the GEPIA portal (http://gepia.cancer-pku.cn, accessed on 10 July 2023) indicates that considering the highest and lowest quartiles of expression, higher *CXCL1* expression in AML cells is associated with poorer prognoses [9,52]. Additionally, higher *CXCL1* expression in the bone marrow of AML patients is associated with worse prognoses [54].

The expression level of other CXCR2 ligands is also linked to prognoses. Specifically, higher *CXCL2* expression in AML cells is associated with poorer prognoses [55,56]. According to the GEPIA portal (http://gepia.cancer-pku.cn, accessed on 10 July 2023), considering the highest and lowest quartiles of expression, higher *CXCL2* expression in AML cells is related to a tendency (*p* = 0.055) for poorer prognoses [9]. Moreover, higher *CXCL2* expression in AML cells in adult patients with cytogenetically normal AML-M5 is associated with poorer prognoses [43,44]. Additionally, higher CXCL2 expression in the bone marrow is related to a tendency (*p* = 0.055) for poorer prognoses in AML patients [54].

According to the GEPIA portal (http://gepia.cancer-pku.cn, accessed on 10 July 2023), higher expression of *CXCL3*, *CXCL5*, and *PPBP* in AML cells is associated with poorer prognoses for patients with the described leukemia [9,52]. Other bioinformatic analyses also confirmed that higher *CXCL5* expression in AML cells is related to poorer prognoses [57]. According to the UALCAN portal (https://ualcan.path.uab.edu, accessed on 10 July 2023), there is a tendency (*p* = 0.093) for poorer prognoses with higher *PPBP* expression in AML cells [8,10].

Regarding CXCL8, higher expression of this chemokine in AML cells is associated with poorer prognoses, but only in cases of AML without FAB M3 [47].

Another CXCR2 ligand that is not a chemokine is *MIF*. Higher expression of *MIF* in AML cells is related to poorer prognoses [38] or tendency (*p* = 0.053) for poorer prognoses in AML patients [8,10].

### 2.6. The Association of CXCR2 Ligands with the Remaining Clinical Characteristics of Patients with AML

In addition to the strong association between the expression levels of CXCR2 ligands in AML cells and prognoses, similar relationships are observed with other clinical aspects. Higher *CXCL8* expression in AML cells is also associated with chemotherapy failure and a high likelihood of recurrence after chemotherapy [34]. Furthermore, lower CXCL8 levels in the blood of adult AML patients are associated with a higher probability of developing graft-versus-host disease (GVHD) after bone marrow transplantation [58].

### 2.7. Mechanisms Regulating the Production of CXCR2 Ligands in the Bone Marrow of Patients with AML

The gathered data indicate a significant association between CXCR2 ligands and the prognosis of AML patients. Therefore, it can be inferred that CXCR2 ligands are involved in the tumorigenic processes in AML. They may act in the bone marrow, where their expression and levels are elevated in AML patients compared to healthy individuals [39]. The higher level of CXCL8 in the bone marrow of AML patients may originate from AML cells themselves [40] as well as from mesenchymal stromal cells (MSCs) [59], with MSCs likely being the main source [59]. Hypoxia may be a factor that increases the expression of CXCR2 ligands in the bone marrow. Under such conditions, CXCL8 production in AML cells is increased [47,60]. Hypoxia in the bone marrow also upregulates MIF expression in AML cells [61], a process dependent on hypoxia-inducible factor-1 (HIF-1). Moreover, hypoxia increases the expression of CXCR2 on AML cells [56], leading to enhanced action of CXCR2 ligands on AML cells in the bone marrow.

Another factor that may increase CXCL8 production in the bone marrow is R-2-hydroxyglutarate, but only in cases of AML with mutations in the isocitrate dehydrogenase (*IDH*)1 or *IDH2* genes [62]. Mutations in these genes result in the production of an oncometabolite, R-2-hydroxyglutarate, in AML cells. This compound induces NF-κB activation in bone marrow stromal cells, leading to increased CXCL8 production in these cells.

Interaction between AML cells and non-leukemic cells in the bone marrow may also contribute to elevated levels of CXCR2 ligands. AML cells secrete CXCR2 ligands, especially CXCL8, which promotes the migration of MSCs to these cells [47]. As a result of the interaction between AML cells and MSCs, the production of CXCL1, CXCL5, CXCL8 [41,59,63], CXCL3, and CXCL6 [64] is increased. This has been shown in co-culture experiments with these two types of cells.

MIF, which is produced by AML cells [61,65], may be responsible for increased CXCL8 production in MSCs. Interaction between AML cells and osteoblasts also leads to increased CXCL8 production [66]. Additionally, exosomes released by AML cells can increase CXCL8 production in bone marrow stromal cells [67]. All these factors contribute to increased production and levels of CXCR2 ligands in the bone marrow. In the bone marrow, CXCR2 ligands participate in tumorigenic processes in AML.

### 2.8. The Significance of CXCR2 Ligands in the Angiogenesis in the Bone Marrow of Patients with AML

AML patients exhibit increased microvessel density in the bone marrow compared to healthy individuals [68,69]. After successful AML treatment, the number of vessels in the bone marrow returns to the levels observed in healthy individuals. Pro-angiogenic factors, particularly those secreted by AML cells, are responsible for the increased number of vessels in the bone marrow of AML patients [69]. AML cells secrete various factors, including vascular endothelial growth factor (VEGF), hepatocyte growth factor (HGF), basic fibroblast growth factor (bFGF), matrix metalloproteinase (MMP)2, and MMP9 [69,70,71].

CXCR2 ligands may also be among these factors in AML patients. Studies in adults [72] and pediatric AML patients [45] confirm this. Among all CXCR2 ligands, CXCL8 may play a key role in the increased microvessel density in the bone marrow of AML patients. However, it should be noted that CXCR2 ligands are not the only factors responsible for increased angiogenesis in the bone marrow of AML patients.

### 2.9. The Significance of CXCR2 Ligands in the Proliferation of AML Cells

CXCR2 ligands may directly affect AML cells and, thus, influence their proliferation. However, only in some patients do CXCR2 ligands increase or decrease the proliferation of AML cells [40]. In most cases, CXCR2 ligands do not affect the proliferation of AML cells.

### 2.10. The Significance of CXCR2 Ligands in the Formation of Extramedullary AML

Another role of CXCR2 ligands in AML-related tumorigenic processes and their impact on prognosis may involve the participation of these chemokines in the migration of AML cells to different organs. Approximately 14% of AML patients experience extramedullary AML, where AML cells infiltrate tissues other than the bone marrow and blood, such as the skin, central nervous system (CNS), and others [73].

Patients with extramedullary AML exhibit higher CXCR2 expression on AML cells than patients with AML cells found only in the blood and bone marrow [52]. This suggests the involvement of the CXCR2 axis in the infiltration of different tissues by AML cells. However, the precise mechanism of CXCR2 involvement in the development of extramedullary AML has not been elucidated. The expression of CXCR2 ligands in the skin and CNS is low compared to other tissues [74], indicating the existence of other mechanisms responsible for the development of extramedullary AML involving CXCR2.

### 2.11. CXCR2 Ligands Induce Chemoresistance in AML Cells

Another possible mechanism of action of CXCR2 ligands in prognosis is their role in causing chemoresistance to anti-leukemic drugs (Figure 1). Studies have demonstrated that CXCR2 ligands, particularly CXCL8, are associated with resistance to cytarabine [37], etoposide [67], and daunorubicin [75].

Exposure of AML cells to daunorubicin leads to increased expression of histone deacetylase 8 (HDAC8) in these cells. Consequently, this results in enhanced activation of NF-κB and increased expression of genes dependent on this transcription factor, including CXCL8. This chemokine then leads to daunorubicin resistance [75].

Additionally, CXCL8 in the bone marrow causes etoposide resistance in AML cells [67]. Moreover, a higher level of CXCL8 in the blood of AML patients is indicative of etoposide resistance [67].

Furthermore, CXCR2 ligands induce resistance to FLT3 tyrosine kinase inhibitors. CXCR2 ligands, including MIF, increase the survival of AML cells exposed to FLT3 tyrosine kinase inhibitors such as gilteritinib [76]. Notably, gilteritinib itself activates NF-κB2, leading to increased MIF expression in AML cells. Subsequently, MIF enhances the expression of CXCR2 and various CXCR2 ligands in AML cells, including CXCL1, CXCL5, and CXCL8 [76]. Activation of this axis results in resistance to FLT3 tyrosine kinase inhibitors. Blocking CXCR2 activity using an inhibitor increases the susceptibility of AML cells to FLT3 tyrosine kinase inhibitors, potentially offering a promising therapeutic approach in treating AML with *FLT3* gene mutations.

### 2.12. Drugs Targeting CXCR2 as Anti-Leukemic Agents

The association between higher expression of CXCR2 ligands and CXCR1 and CXCR2 receptors on AML cells and poorer prognosis suggests a potential therapeutic target in anti-leukemic therapy (Table 2 and Table 3). A study in mice engrafted with U937 cells indicates that blocking CXCR2 activity in these cells increases the survival of these laboratory animals [42]. Moreover, the use of CXCR2 inhibitors, such as SB225002, may enhance the effectiveness of FLT3 tyrosine kinase inhibitors. An in vitro study on the MV4–11 cell line confirmed this finding [76]. It should be noted that SB225002 may act independently of CXCR2. This compound can destabilize microtubules, which may account for its anti-cancer properties [77]. However, research on the implementation of CXCR2 inhibitors in AML therapy is still in its early stages.

## 3. CXCR3 Ligands

### 3.1. Basic Information about CXCR3 and Its Ligands

Regarding CXCR3 ligands, CXCR3 (CD183) exists in three alternative splice variants: CXCR3A, CXCR3B, and CXCR3alt [78]. CXCR3 activation leads to different signal transduction pathways depending on the alternative splice variants. The ligands for CXCR3 include PF-4 (also known as CXCL4), CXCL9 (also known as monokine induced by interferon (MIG)), CXCL10 (also known as interferon-γ-inducible protein 10 (IP-10)), and CXCL11 (also known as interferon-inducible T cell a-chemoattractant (I-TAC)) [7,14]. CXCL9, CXCL10, and CXCL11 can activate all variants of CXCR3, while PF-4 can only activate CXCR3B [78,79].

CXCR3 is expressed on NK cells and T cells [80], making CXCR3 ligands significant in the function of these cells. Furthermore, the CXCR3 axis is also crucial in AML tumorigenic processes. Another property of the discussed axis is the inhibition of angiogenesis [26].

### 3.2. Expression of CXCR3 in AML Cells

The expression level of CXCR3 on AML cells does not differ from the expression level of this receptor on bone marrow CD34^+^ cells [49]. The expression of CXCR3 is highest in AML cells with the FAB M3 and M7 phenotypes [8,10,51]. The expression of CXCR3 is lower in AML cells with FLT3 or nucleophosmin 1 (NPM1) mutations compared to AML cells without these mutations [51].

Additionally, the expression level of CXCR3A on AML cells may depend on other factors. CD34^+^ AML cells have higher CXCR3A expression than CD34^−^ AML cells [40], suggesting that the discussed axis may be significant for AML stem cells.

### 3.3. Expression of CXCR3 Ligands in AML Cells

AML cells in the majority of patients produce PF-4 and CXCL10. Approximately 40% of patients have AML cells that produce detectable levels of CXCL9 and CXCL11 [40]. CXCL9 expression may be highest in AML cells with the FAB M7 phenotype, while CXCL10 and CXCL11 expression is lowest in AML cells with the FAB M3 phenotype [8,10].

The higher expression of CXCR3 ligands in AML cells may be correlated with other chemokines, suggesting a certain AML subtype with specific tumorigenic mechanisms. The expressions of CXCL9, CXCL10, and CXCL11 are correlated with each other and with the expression of CCL5 and CCL23 [40].

### 3.4. Levels of CXCR3 Ligands in Patients with AML

In adult patients with AML, there is a higher level of CXCL10 in the blood compared to healthy individuals [32]. However, another study indicated that elevated levels of CXCL9 and CXCL10 in the blood were observed only in adult patients younger than 50 years old [58,81]. Additionally, research has shown that the level of CXCL10 in the blood of pediatric and adult AML patients is lower than that in healthy individuals and decreases even further after bone marrow transplantation [30].

The blood level of CXCL10 is not associated with FAB classification or FLT3 gene mutation in AML cells [81]. Notably, the lower blood levels of CXCL9 and CXCL10 in adult AML patients may also be associated with the development of GVHD after bone marrow transplantation [58].

In the bone marrow of adult AML patients, there are higher expression and levels of CXCL9 and CXCL10 compared to healthy individuals [39]. The increased CXCL10 level may result from the interaction of AML cells with MSCs [64,81], fibroblasts, and osteoblasts [81]. Co-culture studies of AML cells with these cells have shown that the interaction leads to increased CXCL10 expression. However, another study suggests that the co-culture of AML cells with MSCs only minimally affects the production of CXCR3 ligands or not at all [41].

In contrast, hypoxia may not influence the expression of CXCL9, CXCL10, and CXCL11 in AML cells [60]. Additionally, higher CXCL10 expression in the bone marrow may result from mutations in AML cells in the IDH1 and IDH2 genes, leading to the production of R-2-hydroxyglutarate by these cells. Experiments on StromaNKtert cell lines have shown that R-2-hydroxyglutarate increases CXCL10 expression in bone marrow stromal cells [62].

CXCR3 ligands in the bone marrow can inhibit the proliferation of hematopoietic progenitor cells [82], leading to disrupted hematopoiesis observed in AML patients [83,84]. Similar properties have been demonstrated for CCL3 [85].

### 3.5. The Association of CXCR3 Receptor Expression along with Its Ligands with the Outcomes for Patients with AML

The discussed chemokine axis plays a crucial role in the tumorigenic processes of AML. This is evidenced by the association between the expression level of the CXCR3 receptor and its ligands and the survival of AML patients. Higher expression of CXCR3 [8,10], PF-4 [8,10], CXCL10 [8,9,10,86], and CXCL11 [8,10] in AML cells is associated with poorer prognoses. The mechanisms underlying the involvement of this axis in tumorigenic processes in AML have not been fully elucidated and understood.

### 3.6. The Significance of CXCR3 Ligands in the Proliferation of AML Cells

CXCR3 ligands may increase the proliferation of AML cells in some patients [40,87]. However, these chemokines may also decrease the proliferation of AML cells in the bone marrow in some patients. Furthermore, CXCR2 ligands, particularly PF-4 and CXCL10, in combination with hematopoietic cytokines such as c-kit ligand, granulocyte colony-stimulating factor (G-CSF), granulocyte–macrophage colony-stimulating factor (GM-CSF), and interleukin-3 (IL-3), may reduce the proliferation of AML cells [87,88].

### 3.7. The Significance of CXCR3 Ligands in the Development of Extramedullary AML

In approximately 14% of AML patients, leukemic cells may be present not only in the bone marrow and blood but also in other tissues. The most common location for AML cells besides the bone marrow and blood is the skin [73], referred to as extramedullary AML of the skin. CXCR3 ligands may be responsible for AML cell homing to the skin in adult patients [89]. However, there is low expression of CXCR3 ligands in the skin [74]. Therefore, the exact molecular mechanism responsible for the homing of AML cells with high CXCR3 expression to the skin is not known.

### 3.8. The Association of CXCR3 Ligands with the Condition of AML Patients

CXCR3 ligands not only directly impact the tumorigenic mechanisms in AML but can also influence the patient’s condition. It has been shown that the level of CXCL10 in the blood is correlated with cancer-related fatigue [90]. However, the exact mechanism linking cancer-related fatigue in AML patients with CXCL10 has not been fully understood.

### 3.9. Conclusions

The involvement of the described chemokine axis in the tumorigenic processes of AML has not been well understood (Figure 2). The expression level of the CXCR3 receptor and its ligands in AML cells is closely associated with the prognosis of patients with this leukemia [8,9,10,86]. This indicates that the described axis plays a significant role in the tumorigenic processes of AML, which needs further exploration. Additionally, it is desirable to investigate whether targeting CXCR3 or its ligands has any therapeutic potential in AML patients (Table 4).

## 4. CXCR5 Ligand: CXCL13

CXCL13 is a chemokine that activates CXCR5 (CD185) [91] and CXCR3 (CD183) [92]. Another name for CXCL13 is B-cell-attracting chemokine 1 (BCA-1), and it is associated with important functions for B cells [91,93,94]. CXCL13′s angiogenic properties appear to vary depending on the specific model employed. Research findings suggest that CXCL13 can exhibit either pro-angiogenic or anti-angiogenic characteristics [95,96]. Notably, CXCL13 does not appear to directly influence the proliferation or migration of endothelial cells, as demonstrated in experiments involving human umbilical vein endothelial cells (HUVECs) [96]. However, it is worth noting that CXCL13 can diminish the effects of bFGF on these cells, which could be interpreted as an anti-angiogenic effect. Interestingly, CXCL13 has been observed to facilitate endothelial progenitor cell (EPC) homing, ultimately contributing to angiogenesis, particularly in models related to rheumatoid arthritis [95].

AML cells have higher CXCR5 expression compared to controls [51]. However, AML with FLT3 gene mutation shows lower CXCR5 expression than AML cases without this mutation [51]. The level of CXCR5 expression on AML cells does not impact prognosis in adult patients [8,10,51]. The ligand for this receptor, CXCL13, is produced in small amounts by AML cells in half of the patients [40]. AML cells with the FAB M5 phenotype show the highest CXCL13 expression [8,10]. CXCL13 may promote proliferation in a few patients with AML [40]. However, the lack of associations between CXCL13 and CXCR5 expression levels and prognosis suggests that this axis may not play a significant role in the pathogenesis of AML.

## 5. CXCL14

Chemokine CXCL14 is a chemoattractant for monocytes [97], macrophages [98], B cells [98], and dendritic cells [99]. Additionally, this chemokine may also play a role in activating dendritic cells [100]. Its previous name is breast- and kidney-expressed chemokine (BRAK). The receptor for CXCL14 is not well defined, but it seems that CXCL14 can activate CXCR4, ACKR2 [101], IGF-1R [102], and GPR85 [103]. CXCL14 can also bind to CXCR4 [104]. Interestingly, it appears that CXCL14 may act as a positive allosteric modulator for CXCR4 [105]. Furthermore, CXCL14 exhibits anti-angiogenic properties [106].

The level of CXCL14 expression in AML cells does not correlate with patient outcomes (Table 5) [8,10]. Furthermore, CXCL14 does not influence the proliferation of AML cells [40]. Therefore, it is likely that CXCL14 does not participate in the oncogenic processes in AML and has no clinical significance in this disease.

## 6. Ligand CXCR6: CXCL16

CXCL16 is synthesized as a transmembrane protein [107]. In this form, CXCL16 can act as an adhesion molecule by binding to its receptor CXCR6 (also known as CD186) [108]. The membrane-bound form of CXCL16 can also undergo proteolytic cleavage by a disintegrin and metalloproteinase 10 (ADAM10) [109,110] and ADAM17 [110], resulting in the release of soluble CXCL16, which functions as a chemokine by acting on the CXCR6 receptor. The CXCL16–CXCR6 axis is significant in the functioning of monocytes, macrophages, B cells, CD4+ and CD8+ T cells [111], dendritic cells, NKT cells [112], and NK cells [111]. Additionally, CXCL16 is considered a pro-angiogenic chemokine [113,114].

In adult AML patients, the bone marrow exhibits higher levels of CXCL16 compared to healthy individuals [39]. Bone marrow endothelial cells may be responsible for this, as they show higher CXCL16 expression in AML patients than in healthy individuals [115].

Additionally, AML cells secrete CXCL16, with the lowest expression observed in AML cells with the FAB M3 phenotype [8,10]. Higher CXCL16 expression in AML cells is associated with worse outcomes for AML patients [86], suggesting a significant role for this chemokine in AML tumor progression.

AML cells exhibit higher expression of CXCR6 than control cells [51]. The level of CXCR6 expression on AML cells is associated with FAB subtypes, with AML cells with FAB M0 and M7 phenotypes showing the highest expression [8,10]. AML patients with *FLT3* gene mutation also display lower CXCR6 expression than those without this mutation [8,10,51]. Similar trends are observed in AML cases with *NPM1* gene mutation [51]. Furthermore, higher CXCR6 expression is associated with the medium/high-risk group of AML patients compared to the low-risk group.

The level of CXCR6 expression in the blood of AML patients may be associated with better outcomes [57], but conflicting results have been reported [51].

The significance of the CXCL16–CXCR6 axis in AML oncogenesis is not fully understood. CXCL16 may directly affect AML cells, and in some patients, it may increase AML cell proliferation [40]. However, for the majority of cases, CXCL16 does not influence AML cell proliferation. Exploring the influence of the immune system on AML cells through the CXCL16–CXCR6 axis requires further investigation.

## 7. CXCL17

CXCL17 is a recently discovered chemokine that remains poorly studied. This chemokine can activate GPR35 [116] and promotes the migration of monocytes and dendritic cells [117]. CXCL17 can be considered a pro-angiogenic chemokine as it enhances VEGF expression in macrophages [118]. CXCL17 may also enhance the proliferation and migration of certain tumor cells, such as hepatocellular carcinoma [119] and breast cancer [120].

Currently, the role of CXCL17 and GPR35 in AML has not been thoroughly investigated. Data from the UALCAN portal (https://ualcan.path.uab.edu, accessed on 25 June 2023) showed that the expression levels of CXCL17 and GPR35 in AML cells do not influence patient outcomes [8,10]. However, a trend (*p* = 0.058) towards worse outcomes was observed with higher GPR35 expression on AML cells. On the GEPIA portal (http://gepia.cancer-pku.cn/index.html, accessed on 25 June 2023), higher GPR35 expression on AML cells was associated with poorer patient outcomes [9]. The portal’s data also showed that GPR35 expression is higher on AML cells compared to control cells [9]. According to UALCAN, GPR35 expression is highest in AML cells with the FAB M4–M5 phenotype, while CXCL17 expression is highest in AML cells with the FAB M0–M1 phenotype [8,10].

The CXCL17–GPR35 axis has not been thoroughly studied in AML (Table 6). The impact of GPR35 expression on patient outcomes suggests that this receptor may play a role in AML tumorigenesis. However, GPR35 can also be activated by other substances, such as kynurenic acid [121]. Therefore, the significance of CXCL17 in AML tumorigenesis remains uncertain and requires further investigation.

## 8. Conclusions

The best-characterized α-chemokine in the context of AML tumorigenesis is CXCL12. Consequently, current investigations are focusing on anti-leukemic drugs that target its receptor, CXCR4. The significance of CXCR2 and CXCR3 ligands is also well understood, as their expressions in AML cells are closely associated with patient outcomes. However, there is currently a lack of research on drugs targeting these two axes (ligands–CXCR2 and ligands–CXCR3) in AML therapy. This should be a direction of clinical research in the near future.

Regarding other α-chemokines, CXCL16 appears promising from a clinical perspective. Elevated expression of this chemokine in AML cells is strongly correlated with poorer patient prognoses. Therefore, it is essential to investigate how CXCL16 participates in AML tumorigenesis. Additionally, evaluating the potential therapeutic benefits of targeting CXCL16 in AML patients should be a priority. Developing new drugs that target the axes of ligands–CXCR2, ligands–CXCR3, and CXCL16–CXCR6 may significantly improve the outcomes for AML patients.

## Figures and Tables

**Figure 1 cancers-15-04555-f001:**
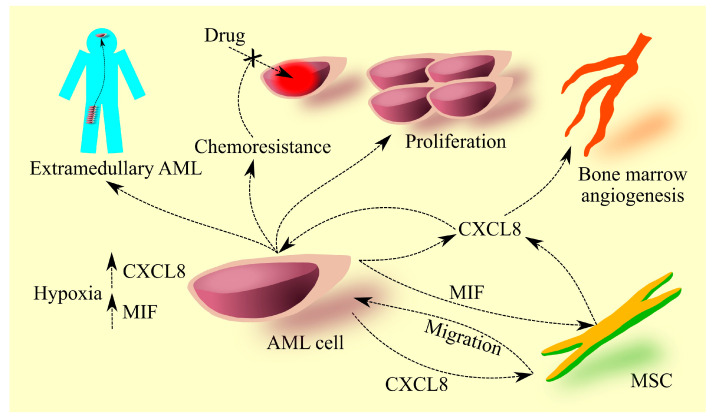
Significance of CXCL8 in AML. The chemokine CXCL8 is secreted by AML cells, and in the bone marrow, hypoxia increases the expression of this chemokine in AML cells. CXCL8 causes migration of MSCs to AML cells, facilitating intercellular communication between them. MIF, secreted by AML cells, increases the expression of CXCL8 in MSCs in the bone marrow. CXCL8 induces angiogenesis in the bone marrow, leading to increased microvessel density in the bone marrow of patients with AML. It should be noted that this process is not solely dependent on CXCL8 but also on other pro-angiogenic factors. Additionally, CXCL8 in the bone marrow acts on AML cells; in some cases, it increases the proliferation of AML cells. CXCL8 also causes chemoresistance and participates in the development of extramedullary AML.

**Figure 2 cancers-15-04555-f002:**
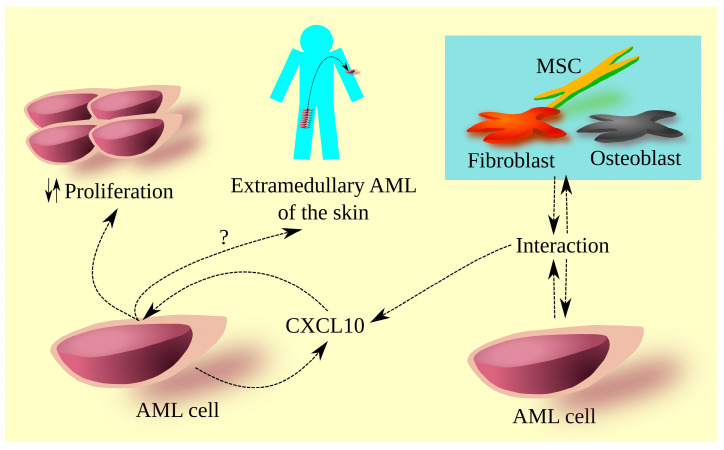
The significance of CXCL10 in AML-related processes. In the bone marrow of AML patients, there is a higher level of CXCL10 compared to healthy individuals. This is associated with CXCL10 production by AML cells. The source of this chemokine in the bone marrow may also depend on interactions between AML cells and other cells such as MSCs, fibroblasts, and osteoblasts. CXCL10 in the bone marrow affects AML cells, leading to changes in the intensity of their proliferation. Depending on the research model, CXCL10 can either increase or decrease the proliferation intensity of AML cells. CXCR3 is also associated with extramedullary AML of the skin; however, the exact molecular mechanism of AML cell homing to the skin is not precisely known.

**Table 1 cancers-15-04555-t001:** Basic information on CXC chemokines (α-chemokines).

Receptor	Ligands	Notes, Properties
**CXCR1 (CD181)**	CXCL6, CXCL8, at high concentrations also CXCL1, CXCL2, CXCL3, CXCL5, CXCL7	The axis is significant in infiltration by neutrophils
**CXCR2 (CD182)**	CXCL1, CXCL2, CXCL3, CXCL5, CXCL6, CXCL7, CXCL8, MIF	Pro-angiogenic properties; axis significant in infiltration by neutrophils
**CXCR3 (CD183)**	PF-4, CXCL9, CXCL10, CXCL11	Anti-angiogenic properties; the crucial axis in CD4+ and CD8+ T cell infiltration; CXCR3 exists in three isoforms generated by alternative splicing—CXCR3A, CXCR3B, and CXCR3alt. All CXCR3 ligands activate all three isoforms of this receptor except for platelet factor-4 (PF-4), which activates only CXCR3B
**CXCR4 (CD184)**	CXCL12	Pro-angiogenic properties; the crucial axis in the functioning of bone marrow
**CXCR3 (CD183), CXCR5 (CD185)**	CXCL13	The crucial axis in the functioning of B cells
**CXCR4 (CD184), atypical chemokine receptor 2 (ACKR2), G protein-coupled receptor (GPR)85,** **insulin-like growth factor-1 receptor (IGF-1R)**	CXCL14	Anti-angiogenic properties. The chemokine may be crucial for B cells, macrophages, and dendritic cells. Positive allosteric modulator for CXCR4
**CXCR6 (CD186)**	CXCL16	Pro-angiogenic properties. CXCL16 exists in two forms: membrane-bound CXCL16 and soluble CXCL16 released by proteases. The membrane-bound form of CXCL16 can bind to CXCR6, activating CXCR6 and promoting cell adhesion. The crucial axis in the functioning of monocytes, macrophages, B cells, CD4+ and CD8+ T cells, dendritic cells, natural killer T (NKT) cells, and natural killer (NK) cells
**GPR35**	CXCL17	Pro-angiogenic properties. The crucial chemokine in the functioning of monocytes and dendritic cells
**CXCR7**	CXCL11, CXCL12	CXCR7 forms a heterodimer with CXCR4, thus functioning together with the CXCR4–CXCL12 axis

**Table 2 cancers-15-04555-t002:** Description of CXCR2 ligands in AML.

Trait/Ligand CXCR2 Analyzed	CXCL1	CXCL2	CXCL3	CXCL5
**Expression levels in patients with AML**	Higher blood levels in AML patients. Higher levels in the bone marrow	Higher expression in bone marrow	Higher expression in bone marrow	No available studies
**Expression in AML cells**	In 1/3 of AML patients, AML cells secrete large amounts of CXCL1. However, another study indicates that the expression of this chemokine in AML cells is low	The expression of CXCL2 in AML cells may be low compared to the expression of other CXCR2 ligands	The expression of CXCL3 in AML cells may be low compared to the expression of other CXCR2 ligands	In 1/3 of AML patients, AML cells secrete large amounts of CXCL5. However, another study indicates that the expression of this chemokine in AML cells is low
**Expression level due to FAB classification**	Independent of FAB classification	Independent of FAB classification	Expression in AML cells with FAB M0–M2 phenotype is higher than in AML cells with FAB M4–M5 phenotype	Independent of FAB classification
**Dependence of expression level on a given mutation**				
**Impact on prognosis**	Higher expression in AML cells is associated with a worse prognosis	Higher expression in AML cells is associated with a poorer prognosis. Notably, higher *CXCL2* expression in AML cells in adult patients with cytogenetically normal AML-M5 is associated with poorer prognoses	Higher expression in AML cells is associated with a worse prognosis	Higher expression in AML cells is associated with a worse prognosis
**Induction of chemoresistance**	The chemokine causes resistance of AML cells to gilteritinib	No data	No data	The chemokine causes resistance of AML cells to gilteritinib
**Association with extramedullary AML**	No data	No data	No data	No data
**Effects on bone marrow microvessel density**	No data	No data	No data	No data
**Trait/ligand CXCR2 analyzed**	**CXCL6**	**PPBP**	**CXCL8**	**MIF**
**Expression levels in patients with AML**	No available studies	No available studies	Higher blood levels in AML patients, especially those younger than 65. Higher levels in the bone marrow	Higher levels in the blood, and higher expression in bone marrow relative to healthy individuals
**Expression in AML cells**	In half of AML patients, AML cells produce low amounts of CXCL6	High expression	In most patients, AML cells produce large amounts of CXCL8	High expression
**Expression level due to FAB classification**	Independent of FAB classification	The highest expression in AML cells with FAB M7 phenotype	Depending on the study, the highest expression in AML with FAB M0 phenotype, lowest in FAB M5, or expression level does not differ by the FAB classification	Lowest in AML with FAB M0 phenotype, highest in AML with FAB M7 phenotype
**Dependence of expression level on a given mutation**			Higher expression with *FLT3-ITD* mutation. Higher expression at translocation t(8;16)(p11;p13) with presence of *MOZ-CBP* fusion gene	
**Impact on prognosis**	No studies available on the association of the expression of this chemokine with prognosis	Higher expression in AML cells is associated with a worse prognosis	Higher expression of this chemokine in AML cells is associated with poorer prognoses, but only in cases of AML without FAB M3	Higher expression in AML cells is associated with a worse prognosis
**Induction of chemoresistance**	No data	No data	The chemokine induces resistance in AML cells to cytarabine, etoposide, gilteritinib, and daunorubicin. Higher level of CXCL8 in the blood of AML patients is indicative of etoposide resistance	The chemokine causes resistance of AML cells to gilteritinib
**Association with extramedullary AML**	No data	No data	No data	No data
**Effects on bone marrow microvessel density**	No data	No data	Association of CXCL8 with angiogenesis in the bone marrow of patients with AML	No data

**Table 3 cancers-15-04555-t003:** Description of CXCR1 and CXCR2 receptors in AML.

Trait/Receptor Analyzed	CXCR1	CXCR2
**Expression in AML cells**	Very low compared to other chemokine receptors.	High compared to other chemokine receptors. CXCR2 expression does not differ between CD34+ AML cells and CD34− AML cells.
**Expression level due to FAB classification**	The highest expression is in AML cells with the FAB M5 phenotype.	The highest expression is in AML cells with the FAB M4–M5 phenotype, while the lowest is in the FAB M3 phenotype.
**Dependence of expression level on a given mutation**	Higher in medium/high-risk group patients than in the low-risk group.	Higher in medium/high-risk group patients compared to the low-risk group. Higher in AML cells with FLT3 gene mutations.
**Impact on prognosis**	Higher expression in AML cells is associated with poorer prognoses.	Higher expression in AML cells is associated with poorer prognoses.
**Induction of chemoresistance**	CXCL8, when acting on AML, induces resistance to etoposide and daunorubicin, but it is not known which receptor is responsible for this property.	Activation of AML cells induces resistance to cytarabine and gilteritinib. CXCL8, when acting on AML, induces resistance to etoposide and daunorubicin, but it is not known which receptor is responsible for this property.
**Association with extramedullary AML**	No data.	Higher expression on AML cells is associated with a higher likelihood of extramedullary AML.
**Effects on bone marrow microvessel density**	A significant association between CXCL8 and bone marrow microvessel density in patients with AML has been demonstrated. However, it is not known which CXCL8 receptor is responsible for this property.	

**Table 4 cancers-15-04555-t004:** Description of CXCR3 and ligands of this receptor in AML.

Trait	PF-4	CXCL9	CXCL10	CXCL11	CXCR3
**Expression levels in patients with AML**	No available studies	Higher blood levels than in healthy individuals, especially those younger than 50. In the bone marrow of AML patients, levels elevated	Higher blood levels than in healthy people, especially those younger than 50. Other studies indicate that levels are lower than in healthy people. In the bone marrow of AML patients, levels are elevated	No available studies	
**Expression in AML cells**	AML cells in the majority of patients produce PF-4	AML cells in approximately 40% of patients produce detectable levels of CXCL9	AML cells in the majority of patients produce CXCL10	AML cells in approximately 40% of patients produce detectable levels of CXCL11	No differences between AML cells and bone marrow CD34^+^ cells
**Expression level due to FAB classification**	The expression does not depend on FAB classification	Highest in AML cells with the FAB M7 phenotype	Lowest in AML cells with the FAB M3 phenotype	Lowest in AML cells with the FAB M3 phenotype	The expression of CXCR3 is highest in AML cells with the FAB M3 and M7 phenotypes
**Dependence of expression level on a given mutation**					Lower expression with a mutation in *FLT3* and *NPM1* genes
**Impact on prognosis**	Worse prognosis with higher expression in AML	No link between expression and prognosis	Worse prognosis with higher expression in AML	Worse prognosis with higher expression in AML	Worse prognosis with higher expression in AML
**Induction of chemoresistance**	No data	No data	No data	No data	No data
**Association with extramedullary AML**	The expression of the CXCR3 receptor is associated with extramedullary AML in the skin. However, there is low expression of CXCR3 ligands in the skin. The molecular mechanism of AML cell homing with high CXCR3 expression in the skin remains unknown				Higher expression of CXCR3 on AML cells is associated with a greater likelihood of extramedullary AML in the skin
**Effects on bone marrow microvessel density**	No data available. The axis exhibits anti-angiogenic properties; however, it is not known whether it plays a role in bone marrow angiogenesis in patients with AML				

**Table 5 cancers-15-04555-t005:** Description of CXCL13, CXCL14, and CXCR5 in AML.

Trait	CXCL13	CXCR5	CXCL14
**Expression in AML cells**	CXCL13 is produced in small amounts by AML cells in half of the patients	Higher expression in AML cells compared to the control	
**Expression level due to FAB classification**	The highest expression is observed in AML cells with the FAB M5 phenotype	The expression level is independent of FAB classification	The expression level is not dependent on FAB classification
**Dependence of expression level on a given mutation**		AML with FLT3 gene mutation shows lower CXCR5 expression	
**Impact on prognosis**	Without an impact on prognoses	Without an impact on prognoses	Without an impact on prognoses
**Induction of chemoresistance**	No available data on the association	No available data on the association	No available data on the association
**Association with extramedullary AML**	No available data on the association	No available data on the association	No available data on the association
**Effects on bone marrow microvessel density**	No available data on the association	No available data on the association	No available data on the association

**Table 6 cancers-15-04555-t006:** Description of CXCL16, CXCL17, and CXCR6 in AML.

Trait	CXCL16	CXCR6	CXCL17	GPR35
**Expression in AML cells I poziom chemokin u pacjentów z AML**	In the bone marrow, there is a higher level of CXCL17 compared to healthy individuals	Higher expression in AML cells compared to the control	No available data	Higher expression in AML cells compared to the control
**Expression level due to FAB classification**	The lowest expression is observed in AML cells with the FAB M3 phenotype	AML cells with FAB M0 and M7 phenotypes show the highest expression	The highest in AML cells with FAB M0–M1 phenotype	Highest in AML cells with FAB M4–M5 phenotype
**Dependence of expression level on a given mutation**		AML cells with FLT3 or NPM1 gene mutations exhibit lower CXCR6 expression. Moreover, higher CXCR6 expression is associated with the medium/high-risk group of AML patients		
**Impact on prognosis**	Higher expression in AML cells is associated with poorer prognoses	Higher expression in the blood is associated with better prognoses. However, other studies have not confirmed this	Without an impact on prognoses	Higher expression in AML cells is associated with poorer prognoses
**Induction of chemoresistance**	No available data on the association	No available data on the association	No available data on the association	No available data on the association
**Association with extramedullary AML**	No available data on the association	No available data on the association	No available data on the association	No available data on the association
**Effects on bone marrow microvessel density**	No available data on the association	No available data on the association	No available data on the association	No available data on the association

## Data Availability

Not applicable.

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
