# Peer review of "The Role of CXCR1, CXCR2, CXCR3, CXCR5, and CXCR6 Ligands in Molecular Cancer Processes and Clinical Aspects of Acute Myeloid Leukemia (AML)"

_cancers, 2023, doi:10.3390/cancers15184555_

Round 1

Reviewer 1 Report

The review article Titled “ The role of CXCR1, CXCR2, CXCR3, CXCR5, and CXCR6 lig-2 ands in molecular cancer processes and clinical aspects of acute 3 myeloid leukemia (AML)” is describing about and significance of CXC chemokine, in AML oncogenic processes.  Authors explored the roles of all CXC chemokines, in particular CXCL1 (Gro-α), CXCL8 (IL-8), CXCL10 (IP-10), and CXCL11 (I-TAC) in AML tumor processes, including their impact on AML cell proliferation, bone marrow angiogenesis, interaction with non-leukemic cells like MSCs and osteoblasts, and their clinical relevance. In this review article authors described how CXC chemokines influence prognosis, association with extramedullary AML, induction of chemoresistance, effects on bone marrow microvessel density, and their connection to FAB phenotype and FLT3 gene mutations. The point-wise comments are as follows;

1.     In this review article, there are floods of CXCR and CXCL and eventually it becomes very confusing, is it ligands or receptors the authors are talking about?

2.     A table should be added that makes clear which is ligand and what is receptors.

3.     In Figure 8 why only CXCL8 is describe not other CXCL8?

Author Response

Rev.1.

  1. In this review article, there are floods of CXCR and CXCL and eventually it becomes very confusing, is it ligands or receptors the authors are talking about?

In our work, we discussed the significance of ligands and receptors. Generally, some of the alpha-chemokine receptors are activated by more than one chemokine. For this reason, in experimental studies, it is easier to demonstrate the importance of the receptor rather than one or a few ligands (statistical significance). Additionally, currently developed drugs target receptors more frequently than ligands. Therefore, we also discussed the importance of receptors. To prepare the reader for the article, we inserted a table describing all CXC chemokines.

  1. A table should be added that makes clear which is ligand and what is receptors.

A table has been added.

  1. In Figure 8 why only CXCL8 is describe not other CXCL8?

All studies on the significance of CXCR2 ligands focus solely on one ligand - CXCL8. In individual works, besides this chemokine, other but not all CXCR2 ligands have been studied. In my latest bioinformatics publication, I have shown that individual CXCR2 ligands differ in their significance in terms of expression regulation. Therefore, very often only certain CXCR2 ligands play a specific role in a given cancer process. For this reason, it was more accurate to mention CXCL8 rather than all CXCR2 ligands, especially since the majority of literature data only investigate CXCL8 and overlook the significance of other CXCR2 ligands. In the text, it has been specified in which AML cancer processes a given CXCR2 ligand (other than CXCL8) plays a specific role in AML.

Reviewer 2 Report

The manuscript presented by Korbecki et al., addresses a significant topic of great interest, however, the way of presenting the information is confusing, they only limit themselves to writing a series of ideas in too long paragraphs, without generating subheadings, especially in chapter 2 and 3, this makes the review quite difficult to read. It is requested to better organize the content of the chapters, perhaps a subheading for each ligand (chapter 2).

Additionally, at the end of the review regarding each of the ligands of the CXCR receptors there should be a partial conclusion mentioning the influence that each ligand ultimately has or the activation of each receptor in "prognosis, association with extramedullary AML, induction of chemoresistance , effects on bone marrow microvessel density, and their connection to FAB phenotype and FLT3 gene mutations" as proposed in the abstract

Being such an extensive review, there should be more than one image that helps to understand the interaction of the ligands with their receptor and its relationship with the clinical aspects of acute myeloid leukemia. Similarly, the list of references is very short and many of the works consulted are other reviews on the same topic.

Author Response

Rev.2.

The manuscript presented by Korbecki et al., addresses a significant topic of great interest, however, the way of presenting the information is confusing, they only limit themselves to writing a series of ideas in too long paragraphs, without generating subheadings, especially in chapter 2 and 3, this makes the review quite difficult to read. It is requested to better organize the content of the chapters, perhaps a subheading for each ligand (chapter 2).

Subsection headings have been added. Overly long paragraphs have been divided. Some sections have been restructured to discuss each ligand sequentially.

Additionally, at the end of the review regarding each of the ligands of the CXCR receptors there should be a partial conclusion mentioning the influence that each ligand ultimately has or the activation of each receptor in "prognosis, association with extramedullary AML, induction of chemoresistance , effects on bone marrow microvessel density, and their connection to FAB phenotype and FLT3 gene mutations" as proposed in the abstract

At the end of each section discussing each axis, summary tables have been inserted covering the entire section. Tables discussing the significance of CXCR1 and CXCR2 receptors have been added. Existing tables have been expanded according to the reviewer's recommendation.

Being such an extensive review, there should be more than one image that helps to understand the interaction of the ligands with their receptor and its relationship with the clinical aspects of acute myeloid leukemia.

A figure discussing the CXCL10-CXCR3 axis has been added.

Similarly, the list of references is very short and many of the works consulted are other reviews on the same topic.

Out of the 120 bibliographic references, only 3 are now reviews, which accounts for just 2.5%. This is significantly below the typical threshold of 20% for review articles in many editorial guidelines. Our review, being the first to cover CXC chemokines other than CXCL12 in AML cancer processes, offers novelty in the realm of review articles.

Round 2

Reviewer 2 Report

Thanks to the authors for addressing the comments made. I have no further comments on this matter or objection to the publication of this manuscript.